# Adherence to self-care practices and associated factors among heart failure patients in Ethiopia: A systematic review and meta-analysis

**Firomsa Bekele** [1]\*, **Lalise Tafese**[2], **Addisalem Workie Demsash**[2], **Hana Tesfaye**[3], **Busha Gamachu Labata** [4], **Ginenus Fekadu** [4,5]

1 Department of Pharmacy, College of Health Science, Mattu University, Mattu, Ethiopia, 2 Department of Health Informatics, College of Health Science, Mattu University, Mattu, Ethiopia, 3 Department of Midwifery, College of Health Science, Mattu University, Mattu, Ethiopia, 4 Department of Pharmacy, College of Health Science, Wallaga University, Nekemte, Ethiopia, 5 School of Pharmacy, Faculty of Medicine, The Chinese University of Hong Kong, Shatin, N.T, Hong Kong, China

\* firomsabekele21@gmail.com

**Data Availability Statement:** The data are only available upon request. The data would be guarded carefully by our third party data for the only purpose of this scientific study. Participants were

## Abstract

### Background

Heart failure is the leading cause of hospital stays, medical expenses, and fatalities, and it is a severe problem for worldwide public health. Successful heart failure therapy requires a high level of self-care as well as devotion to different elements of the treatment plan. Despite the positive effects of heart failure self-care on health outcomes, many heart failure patients engage in insufficient self-care behaviors. Additionally, conflicting information has been found regarding the prevalence and predictors of self-care behaviors in Ethiopia. As a result, this review's objective is to provide an overview of the most recent studies on Ethiopian heart failure patients' self-care practices.

### Methods

We have used four databases such as PubMed, Science Direct, Scopus and Google Scholar. Eventually, the final systematic review and meta-analysis contained eleven papers that matched the eligibility requirements. A systematic data extraction check list was used to extract the data, and STATA version 14 was used for the analysis. Heterogeneity was evaluated using the $I^2$ tests and the Cochrane Q test statistic. To examine publication bias, a funnel plot, Egger's weighted regression, and Begg's test were utilized.

### Result

The pooled magnitude of adherence to self-care was 35.25% (95%CI: 27.36–43.14). The predictors of good adherence to self-care behavior includes heart failure knowledge (odds ratio = 5.26; 95% CI, 3.20–8.65), absence of depressive symptoms (odds ratio = 3.20;95% CI,1.18–8.70), higher level of education (AOR = 3.09;95%CI,1.45–6.61), advanced New York Heart Association (NYHA) class (odds ratio = 2.66; 95% CI, 1.39–5.07), absence of

not signed consent for data publicly. For all these
reasons and following the indications of the
research review committee of College of Health
Sciences, Mettu University, the authors must not
upload the dataset to a stable, public repository.
Interested, qualified researchers can access the
data by requesting our third party, Mattu University
(mattuniversity@meu.edu.et).

**Funding:** The authors received no specific funding
for this work.

**Competing interests:** The authors have declared
that no competing interests exist.

**Abbreviations:** AOR, Adjusted Odds Ratio; CI,
Confidence Interval; HF, Heart failure; NYHA, New
York Heart Association; PRISMA, Preferred
Reporting Items for Systematic Reviews and Meta-
analyses; SNNP, Southern Nation Nationalities and
People.

comorbidity(odds ratio = 2.92; 95% CI,1.69–5.06) and duration of heart failure symptoms
(odds ratio = 0.37; 95% CI, 0.24–0.58).

## Conclusion

The extent of self-care behavior adherence is shown to be low among heart failure patients.
This study showed a positive relationship between self-care behavior and factors such as
proper understanding of heart failure, the absence of co-morbidity, depression, higher levels
of education, a longer duration of heart failure symptoms, and advanced classes of heart
failure disease. Therefore, a continuous health education should be given for patients to
enhance their understanding of heart failure. Besides, special attention should be given for
patients having co-morbidity and depressive symptom.

## Background

Heart failure (HF) is a clinical condition that is indicated by signs and symptoms of fluid over-
load or inadequate tissue perfusion as a result of the heart's decreased cardiac output, which is
required to meet metabolic needs and accommodate venous return [1, 2]. It is a serious public
health problem that is to blame for most global hospital stays, medical costs, and fatalities [1–
3]. The fatality rate for heart failure patients in Africa is three to four times greater than in
Western countries, especially in Sub-Saharan Africa [2].

Despite improvements in pharmaceutical therapy, heart failure morbidity and mortality
remain high. Therefore, non-pharmacological management of heart failure, which primarily
focuses on self-care management, deserves more attention [1, 4–7]. Doctors and other health-
care professionals devote a significant amount of time in each clinical encounter to educate
patients and their families about the requirements of self-care in order to reduce heart failure
exacerbations and re-hospitalizations [8].

Self-care behaviors become essential to halt the development of cardiac remodeling and
avoid rapid decompensation, which can produce subpar clinical outcomes [9]. Self-care is
viewed as the cornerstone of HF therapy and entails essential practices that have been shown
to improve HF clinical outcomes. These include adjusting one's lifestyle by taking prescribed
medications as directed, sodium diet restriction, exercising, reducing liquid intake, and rou-
tinely weighing oneself [2, 10].

A significant amount of self-care and adherence to various components of the treatment
plan are necessary for HF therapy to be successful [10]. Inadequate self-care practices lead to
higher rates of morbidity and mortality, lower quality of life, and higher health care costs
because of more outpatient treatment and higher hospital readmission rates [10–12].

Despite the impact of HF self-care on positive health outcomes, many HF patients don't
practice enough self-care [13]. People with HF who must adhere to a multi-pronged treatment
plan are starting to understand how challenging it is to maintain self-care behavior [10, 11,
14]. Self-care practices were poor, according to several studies conducted in Ethiopia [1, 11,
15–18]. Few studies reportedly found that good self-care behaviors are generally practiced [2,
19, 20].

This lack of compliance may be due to the complexity of such changes, the difficulty of self-
care, the lack of perceived need for self-care, the necessity for long-term behavioral adjust-
ments, a lack of motivation, or any of these factors [21]. In addition, a wide range of factors

may affect how well HF patients follow self-care recommendations. Patients' characteristics like age, sex, marital status, religion, place of residence, level of education, occupation, and family income are among them, as are clinical characteristics like length of diagnosis, stage of HF, co-morbidity, prior hospitalization, awareness of HF, presence of depressive symptoms, and family support [15, 17, 18, 20, 22, 23].

Despite a variable reports of magnitude and predictors of self-care practices in Ethiopia, there was no systematic review and meta-analysis conducted and the purpose of this paper is to summarize the recent findings on factors related to self-care behavior in order to provide an appropriate intervention.

## Methods

### Searching strategy

The objective of the review was to conclude the magnitude and risk factors of adherence to self-care practice among heart failure patients in Ethiopia. The Review protocol was registered on PROSPERO CRD42023423492.

The protocol of PRISMA 2020 was used to undertake this systematic review and meta-analysis [24]. Three data bases like PubMed, Science direct, Scopus and Google scholar were used. The time period used to conduct this review was from the February 9 to March 9, 2023. The last date to search was March 4, 2023. The MESH term for the database is ((Self-care Practice) OR (heart failure)) AND (associated factors)) AND (prevalence)) AND (Ethiopia).

### Data collection process, items and extraction

Three authors namely FB, LT, and AWD were involved in collecting different literatures. Reference management software (endnote version X7.2) used to combine search results from databases and to remove duplicate articles. Data were extracted by two data extractors (FB and LT) using a standardized data extraction checklist on Microsoft excel. For the first outcome (magnitude), the data extraction checklist included author name, year of publication, region, study design, sample size and number of participants with the outcome. For the second outcome (associated factors), data were extracted in a format of two by two tables, and then the log OR for each factor was calculated based on the findings of the original studies. Discrepancies between two independent reviewers were resolved by involving a third and fourth reviewers (ADW and HT) after discussion for possible consensus. BGL and GF have overseen the overall process of data extraction and synthesis. The participant recruitment dates and/or date on which medical records were not accessed since our study was systematic review and meta-analysis.

### Eligibility criteria

The findings published related to magnitude and predictors of adherence to self-care practice among heart failure patients in Ethiopia having all primary outcome and full texts available were included. The articles with unknown primary outcomes, systematic reviews and meta-analysis studies, not peer reviewed and commentary to editors were not eligible. The review used the CoCoPop (condition, Context, and Population) framework to assess the eligibility of the studies.

The study Population (POP) was heart failure patients, the Condition (CO) was adherence to self-care practice, and the context (CO) studies conducted in Ethiopia.

## Outcome measurement

There were two main outcomes. The primary outcome of interest was the prevalence of adherence to self-care practices among heart failure patients, which was estimated as the total number of patients adhered to self-care practices divided by the total number of heart failure patients multiplied by 100. The second outcome was identifying factors associated with an adherence to self-care practice, which were determined using the odds ratio (OR) and calculated based on binary outcomes from the included primary studies.

The extent of self-care practice practices is low if less than the midpoint (50%) of the revised nine-item European Heart Failure Self-care Behavior Scale (EHFScBS-9) [25, 26].

## Quality assessment

The Joanna Briggs institute meta-analysis of statistics assessment and review instrument (JBI-MAStARI) was used for quality assessment [27].

## Data analysis and synthesis

Data exported to STATA V. 14 to calculate the pooled effect size with 95% CIs. To check heterogeneity among the included studies, the Cochran $Q$ test (chi-squared statistic) and $I^2$ statistic on forest plots were computed. Cochran's $Q$ statistical heterogeneity test is considered statistically significant at $P \leq 0.05$. $I^2$ statistics range from 0 to 100% and $I^2$ statistic values of 0, 25, 50, and 75% were considered as no, low, moderate, and high degrees of heterogeneity, respectively. A funnel plot was used to assess publication bias. Asymmetry of the funnel plot is an indicator of publication bias. Besides, Egger's weighted regression and Begg's test were used to check publication bias. Statistical significance of publication bias was declared at a P-value of less than 0.05.

## Risk of bias

The study population was known in all articles. We have obtained complete outcome variables in all articles. In all articles involved, selective reporting and publication bias were not obtained.

## Results

### Search results

A total of 8,950 articles were obtained up on initial searching from PubMed, Science direct, Scopus and Google scholar. A total of 8,006 articles were removed due to duplications. Finally a total of 913 articles were excluded by observing their title and abstracts. Consequently, only 31 articles were subject to a full-text review. Finally, 11 articles were selected to be included in our review [Fig 1].

### Characteristics of included studies

In our systematic review and meta-analysis the filtered articles were cross-sectional studies. The majority of the participant were female in eight of the articles [2, 15–20, 23], whereas male was predominant in the three articles [1, 11, 22]. The total sample was 3,657 heart failure patients, ranging from 235 to 424. Regarding to the study settings, five articles were from Oromia [2, 11, 19, 22, 20] and two were from Amhara [1, 18],two were from Addis Ababa [15, 16], each one article was from SNNP [17] and Tigray [23] [Table 1]. The variations seen in Oromia might be due to a difference in publication years and large variations in sample size.

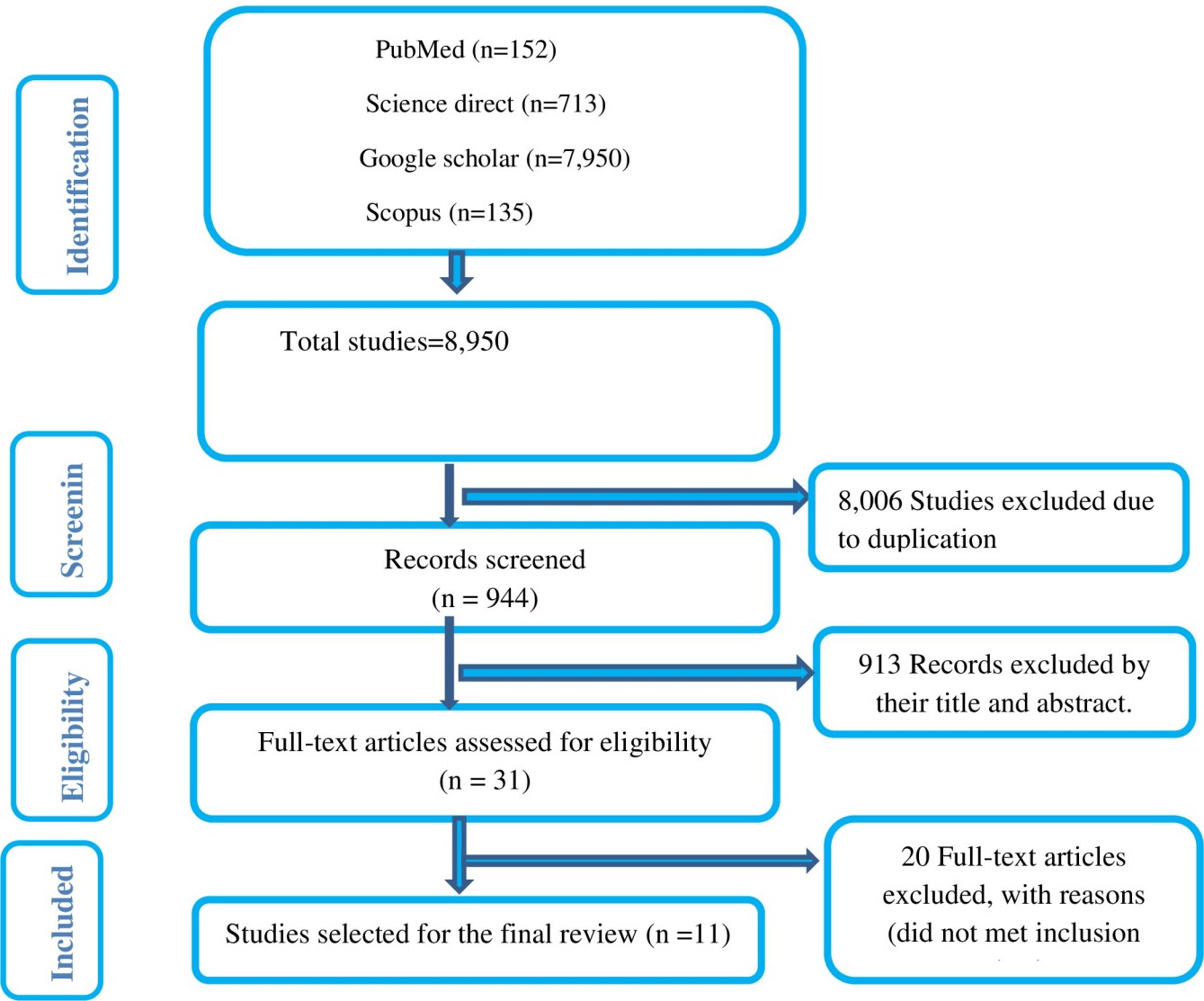

**Fig 1. Flow chart of the systematic research and study selection process.**

### Adherence to self-care practices

The pooled magnitude of adherence to self-care was 35.25% (95%CI: 27.36–43.14). Heterogeneity was not observed across the included studies (I2 = 0.0, p = 0.552). Both the highest (53.6%) [2] and lowest (17.4%) [11] adherence level of self-care practices was reported in Oromia [Fig 2].

### Publication bias

To assess the presence of publication bias, funnel plot, Egger test and Begg's test at 5% significant level were computed. The funnel plot looks asymmetry, but the Egger test and Begg's test showed there is no statistically significant for the presence of publication bias with p-value = 0.363 and 0.175, respectively [Fig 3].

**Table 1. Summary of included studies on self-care practices among heart failure patients in Ethiopia, 2023.**

| Authors | Years of publication | Region | Study design | Sample size | Gender (Male) | Good adherence (95%CI) |
|---|---|---|---|---|---|---|
| Belayneh et al [1] | 2022 | Amhara | Cross-sectional | 312 | 58.6% | 32.9(27.61,38.18) |
| Almaz et al [2] | 2022 | Oromia | Cross-sectional | 420 | 47.1% | 53.6(48.80,58.34) |
| Negese et al [11] | 2015 | Oromia | Cross-sectional | 328 | 55.5% | 17.4%(13.28, 21.48) |
| Getahun et al [19] | 2021 | Oromia | Cross-sectional | 424 | 42.9% | 51.2%(46.42,55.94) |
| Aemiro et al [15] | 2022 | Finfinne | Cross-sectional | 294 | 41.84% | 32.7%(27.29,38.01) |
| Bethlehem et al [16] | 2021 | Finfinne | Cross-sectional | 396 | 48.5% | 28%(23.61,32.45) |
| Enu et al [17] | 2022 | SNNP | Cross-sectional | 235 | 45.4% | 34.1%(27.92,40.2) |
| Jemal et al [22] | 2014 | Oromia | Cross-sectional | 264 | 51% | 40.8%(34.75,46.82) |
| Temesgen et al [20] | 2022 | Oromia | Cross-sectional | 266 | 45.9% | 50.0%(43.99,56.01) |
| Tsegu et al [23] | 2020 | Tigray | Cross-sectional | 408 | 45.1% | 45.8%(41.0,50.67) |
| Mohammed et al [18] | 2019 | Amhara | Cross-sectional | 310 | 35.8% | 22.3%(17.63,26.89) |

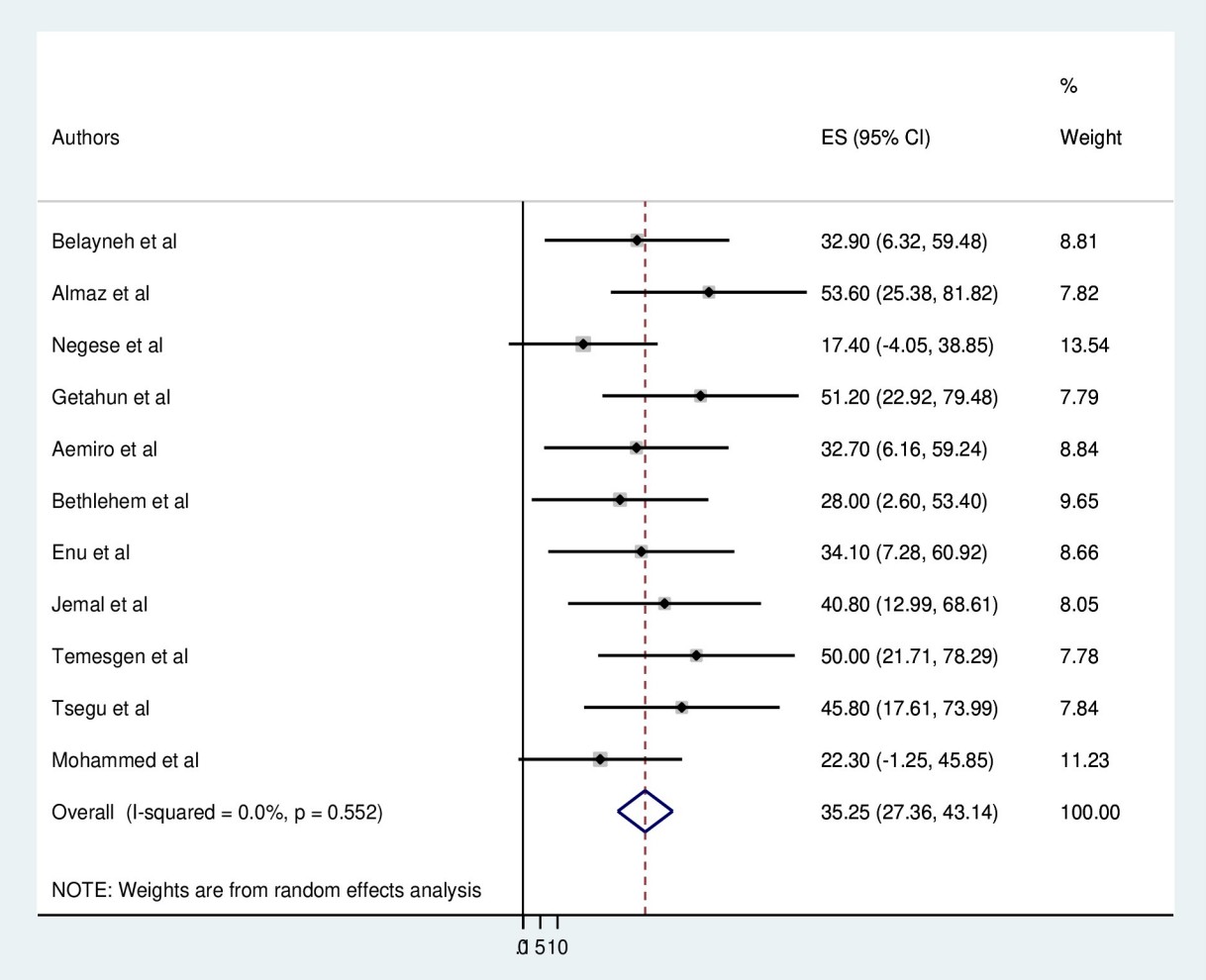

**Fig 2. Forest plot of the pooled magnitude of adherence to self-care practices among heart failure patients in Ethiopia, 2023.**

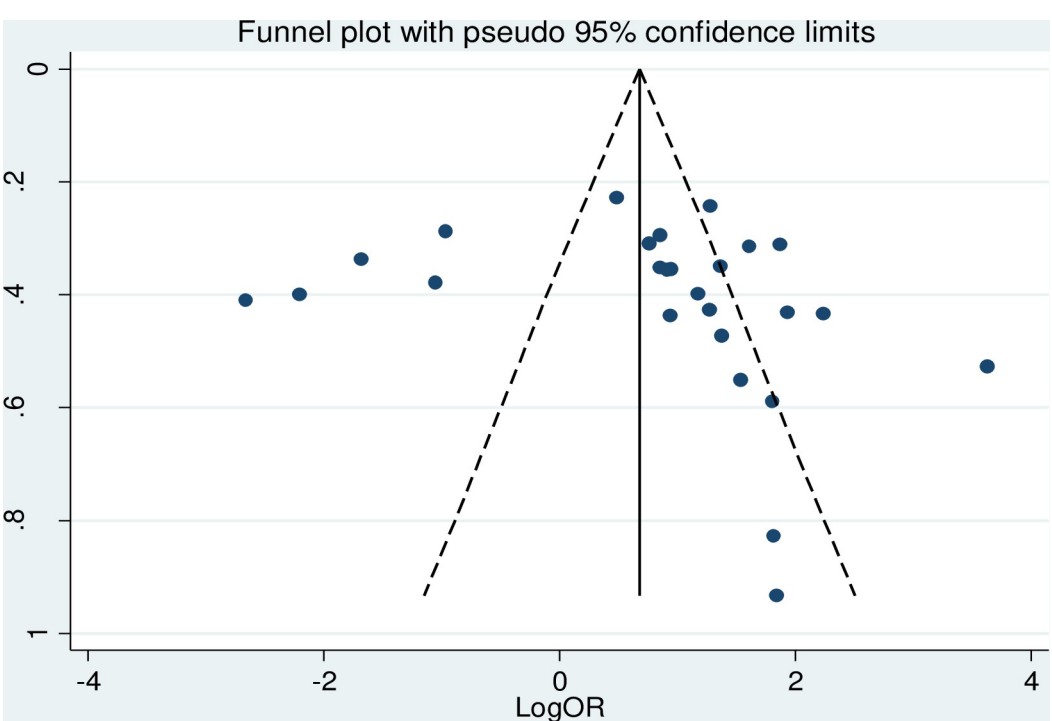

**Fig 3. Funnel plot of the included studies to test publication bias in Ethiopia, 2023.**

### Factors associated with adherence to self-care practices

To identify the pooled predictors of self-care practices, eleven studies were included in the meta-analysis. The pooled effects of odds ratio was assessed by using the command of "metan logor selogor, xlab(0.1,1,10) label(namevar = authors) by (factors)random texts(180) eform." Patients with HF that had a good knowledge on heart failure were 5.26 times more likely to practice good self-care than their counterparts (AOR = 5.26; 95% CI, 3.20–8.65). Similarly, patients hadn't a depressive symptoms were 3.20 times more likely to practice good self-care (AOR = 3.20;95% CI,1.18–8.70). Patient that had a higher level of education were 3.09 times more likely to had a good adherence to self-care behaviors (AOR = 3.09;95%CI,1.45–6.61).

Regarding NYHA classification, those patients with NYHA class III and IV were 2.66 times more likely had good adherence to self-care recommendations compared to NYHA class I and II.(AOR = 2.66; 95% CI, 1.39–5.07). Heart failure patients that had not co-morbid disease were 2.92 times more likely adhered to self-care practices than patients having co-morbidity. (AOR = 2.92; 95% CI,1.69–5.06). However, patients having less than one year duration of heart failure were 63% less likely to adhere to self-care practices. (AOR = 0.37; 95% CI, 0.24–0.58) [Fig 4].

### Discussion

To the best of our knowledge, this meta-analysis and systematic review are the first of its kind that conducted at the national level to estimates magnitude and identifies factors associated with a adherence to self-care practices among heart failure patients in Ethiopia.

The pooled prevalence of adherence to self-care practices among heart failure patients in Ethiopia is 35.25%. In Ethiopia, the prevalence of adherence to self-care practices is relatively increasing from previous studies (17.4% in 2015) to recent ones (53.6% in 2022). The lack of an interventional strategy and ongoing monitoring of heart failure patients may be to blame

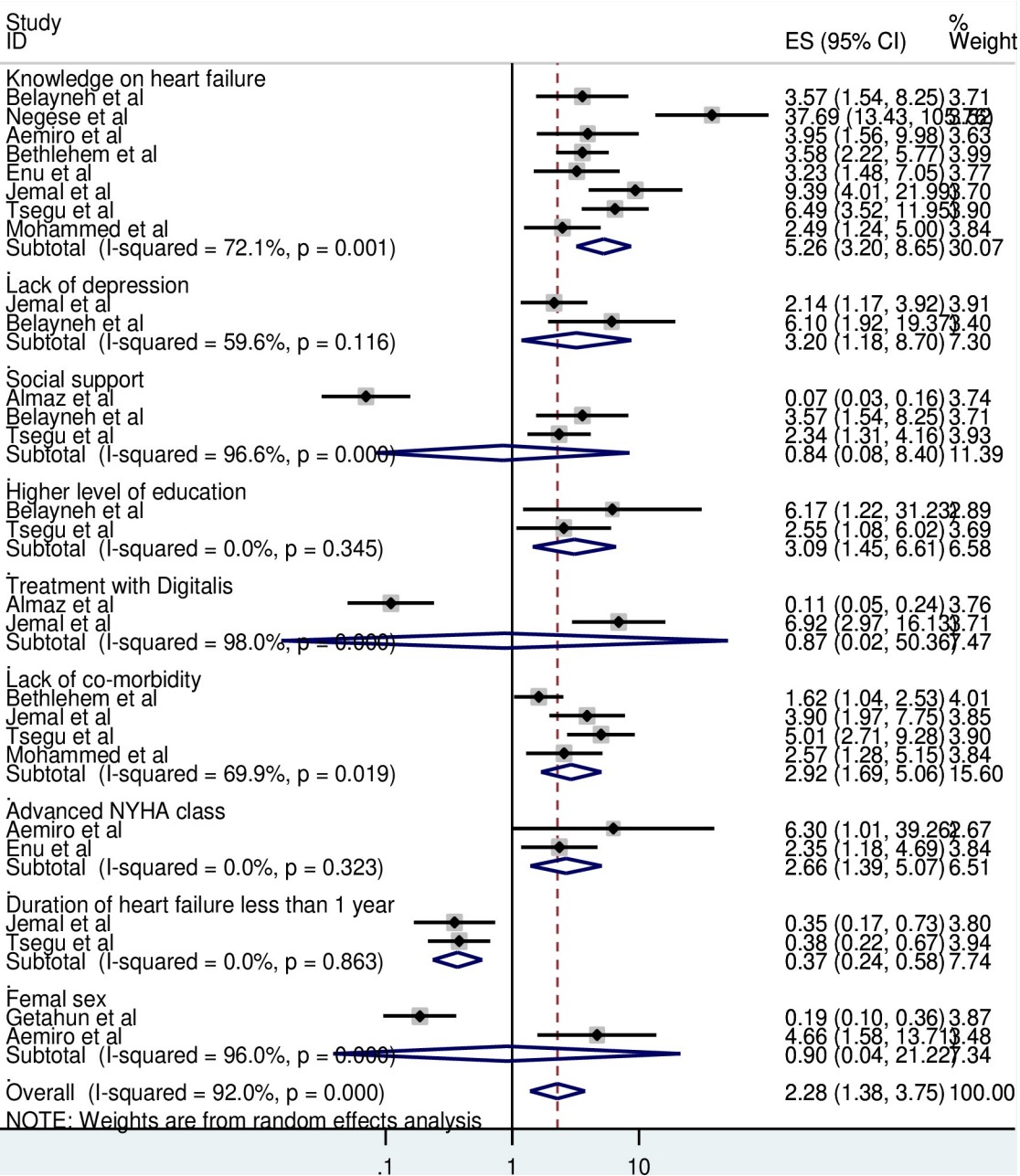

**Fig 4. Forest plot of association factors of self-care practice among heart failure practice in Ethiopia, 2023.**

for the relative increase between earlier trials and more recent ones. The rate of adherence is equivalent to that of the study done in Brazil(35.17%) [9], lower than that of the studies done in the Netherlands(41%) [6] and Taiwan(53.37%) [28], and higher than that of Korea(31.98%) [14]. This discrepancy could be due to the difference in the sample size and sampling techniques and different data collection used.

This systematic review and meta-analysis also identified factors associated with adherence to self-care practices. Heart failure patients who had a good knowledge on heart failure had more likely to have a good adherence. This finding is supported by a study conducted in a California [8] and Korea [14]. This could be explained by the fact that a patient's understanding of

their illness is a requirement for improving self-care behaviors, and that individuals with good knowledge exhibited good health-seeking behaviors to avoid needless readmissions to the hospital.

Heart failure patients who were on advanced stage were more likely to be adhered to self-care practices than their counterparts. This is similar with the systematic review of European Heart Failure Self-Care Behavior Scale studies [13] and inconsistent with Korea [14]. This might be as a result of their frequent interactions with medical personnel and worry of a bad treatment outcome. Additionally, patients with NYHA functional classes I and II think their HF symptoms may have completely subsided more frequently, and this belief may hold true in the future.

Heart failure patients who were having higher level of education were more likely to be adhered to self-care practices than their counterparts. Similar finding was reported according to systematic review of European Heart Failure Self-Care Behavior Scale studies [13] and Nepal [3]. This can be explained by the fact that people with higher educational levels have higher levels of reasoning and decision-making for performing self-care behaviors and they can easily understand the information required for self-care that leads them to good adherence.

Patients were more likely to follow their self-care routines if they did not exhibit depressed symptoms. According to results from Nezerland [12] and a systematic evaluation of research using the European Heart Failure Self-Care Behavior Scale [13], similar findings have been observed. Patients with better mental health are more likely to adhere to self-care recommendations because they have unaltered thinking abilities and a positive attitude toward maintaining their health. Depression may increase the burden of patients' overall clinical condition, making the patients less likely to follow the recommended self-care practice.

Patients were more likely to follow their self-care routines if they did not have a co-morbid ailment. This is consistent with a Japanese study's result that diabetes mellitus is a predictor of poor self-care behaviors [7]. Patients with diabetes mellitus need to engage in self-care activities linked to the disease, such as a special diet and exercise to control blood sugar. Patients with HF who also have diabetes mellitus require a different or more intensive treatment regimen, as well as more complex self-care behavior components. Such patients are likely to have difficulty with self-care behavior, and they may therefore need individually tailored support to help them combine the needed self-care behavior.

Co-morbid participants were more likely to be non-adherent than non-co-morbid participants, which may be because patients with multiple chronic illnesses must overcome additional functional, cognitive, and physical barriers in order to carry out multicomponent self-care recommendations. These patients are more likely to see many doctors, and they may get advice that is unclear or contradictory, which could lower their degree of adherence.

Patients were less likely to adhere to self-care practices if their heart failure symptoms had not been present for a longer period of time. This result was consistent with research conducted in Southern California Ohio [29] and Brazil [30], This is because individuals who were recently diagnosed with HF had a harder time identifying their own HF symptoms. Therefore, compared to newly diagnosed patients, experienced patients were more likely to use appropriate self-care strategies.

## Limitation of the study

As the limitation, the sample size of the included studies was small. In addition, all of the studies included in this review were cross-sectional study design; as a result, the causal effect relationship was could not be identified.

## Conclusion

In current systematic review and meta-analysis, the magnitude of adherence to self-care behavior is among heart failure patients is found to be low. This study showed that adequate knowledge of heart failure, absence of co-morbidity, depression, higher level of education, longer duration of heart failure symptoms, and advanced class of heart failure disease were positively associated with the adherence to self-care behavior. Therefore, a proper health education towards improving the patient's knowledge is vital to improve self-care behavior. Besides, special attention should be given for patients having co-morbidity, advanced stage of heart failure, and shorter duration of heart failure symptom. Finally, it is important for health care providers to learn to recognize depressive symptoms in heart failure patients and treat depressed patients according to existing psychiatric guidelines.

## Supporting information

**S1 Checklist. PRISMA-2020 (Preferred Reporting Items for Systematic Reviews and Meta-Analysis-2020) checklist.**
(DOCX)

**S1 Data. The microsoft excel data of pooled magnitude of adherence to self-care practices.**
(XLSX)

**S2 Data. The microsoft excel data of the pooled associated factors of adherence to self-care practices.**
(XLSX)

## Acknowledgments

We would like to thank all authors of the studies included in this systematic review and meta-analysis.

## Author Contributions

**Conceptualization:** Firomsa Bekele, Hana Tesfaye, Ginenus Fekadu.

**Data curation:** Firomsa Bekele, Hana Tesfaye.

**Formal analysis:** Firomsa Bekele, Lalise Tafese, Addisalem Workie Demsash, Ginenus Fekadu.

**Investigation:** Lalise Tafese.

**Methodology:** Firomsa Bekele.

**Software:** Addisalem Workie Demsash.

**Supervision:** Firomsa Bekele, Addisalem Workie Demsash, Ginenus Fekadu.

**Visualization:** Busha Gamachu Labata, Ginenus Fekadu.

**Writing – original draft:** Firomsa Bekele, Busha Gamachu Labata.

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
