## [Decision Letter · Decision Letter 0]

27 Apr 2023

PONE-D-23-06552Self-care practices and associated factors among heart failure patients in Ethiopia: A systematic review and Meta-AnalysisPLOS ONE

Dear Dr. Bekele,

Thank you for submitting your manuscript to PLOS ONE. After careful consideration, we feel that it has merit but does not fully meet PLOS ONE’s publication criteria as it currently stands. Therefore, we invite you to submit a revised version of the manuscript that addresses the points raised during the review process.

Reviewer #1: Independent Review Report

Evaluation

The evaluation is for the review article titled “Self-care Practices and associated factors among heart failure patients in Ethiopia: A systematic review and Meta-Analysis”

With the present study, the following areas are observed, commented and recommended.

1. Throughout the main text, it was sequential and well organized.

2. It requires grammatical corrections throughout the document.

3. This manuscript has a sum up of novelty.

Comments to the author/s

Abstract:

1. Try to avoid abbreviation of words in the abstract part

2. “We have used three databases such as Pub Med, Science Direct and Google Scholar.

Why do you exclude other important databases that you might get potential primary studies from?

3. Heterogeneity was evaluated using the I2 tests and the Cochrane Q test statistic.

What is the advantage of performing the Q test in addition to the I2 test to assess heterogeneity?

4. To examine publication bias, a funnel plot, Egger's weighted regression, and Begg's test were utilized. What is the importance of using different tests?

5. “The extent of self-care behaviour adherence is shown to be low among heart failure patients”. What is the standard reference to say “low or high”?

Method

6. Please use the uniform reference citation system

7. Please try to strictly adhere to all the PRISMA 2020 guidelines to write up your paper, especially the method section.

8. Does this review registered on PROSPERO?

9. What is your last search date included in the method part?

10. What you have done if a study has low quality or higher risk?

11. “Finfinnee" is not a legal name, not yet known

12. What you have done if there is heterogeneity and publication bias between the primary studies?

13. How to perform the pooled effect of odds ratio to identify the factors associated with good selfie care? Please show on your paper as it is critical.

Result

14. Table 1 column 6 gender (male). What is this???

15. Why you have performed sub-group analysis and sensitivity test?

Reviewer #2: Dear authors, thank you for your contribution. In general the issue you raised is very good. However, I have some comments and questions which described below.

1. Language edition service is required throughout the document

2. Please rephrase this sentence found in your abstract’ ‘The patient's understanding must therefore be improved by effective health education if self-care behaviour is to be improved.’’

3. The databases scrutinized to search studies are too small. Why only three databases? I think some papers are missed. Therefore please search on at least familiar databases like HINARI, Scopus, and AJOL

4. The MESH terms are little, why?

5. Why you preferred CoCoPop?

6. With this visible heterogeneity among studies, it is difficult to accept. Therefore, I need further evidence /explanation for this because the authors even mentioned the presence the presence of heterogeneity between studies conducted in Oromia. Why?

7. In your discussion part you mentioned this’ ’The prevalence of adherence is comparable with the study conducted in Brazil (35.17%)(9) and lower than the study of Netherland(41%)(6) and Taiwan(53.37%)(26) and higher than the study of Korea(31.98%)(14). This discrepancy could be due to the difference in the study method.’’ What type of difference in study method?

8. On your limitation please rephrase this sentence’ ‘The small sample due to a limited number of included studies with a small sample size for all included studies was a limitation of this study. ‘’

9. The number of studies included in your study is 11 but in the funnel plot it is more than 20 (each dot on the figure represent one study), Why this variation

Thank you

We look forward to receiving your revised manuscript.

Kind regards,

Mohammed Feyisso Shaka, MPH

Academic Editor

PLOS ONE

Journal Requirements:

- https://doi.org/10.1186/s12889-019-8133-y

- https://doi.org/10.1016/j.hrtlng.2008.11.002

- https://www.slideshare.net/negesesewagegn/adherence-toselfcarebehavioursandknowledgeontreatmentamongheartfailurepatientsinethiopiathecaseofatertiaryteachinghospital23760419s4001

In your revision ensure you cite all your sources (including your own works), and quote or rephrase any duplicated text outside the methods section. Further consideration is dependent on these concerns being addressed.

"NA"

"No competing interest"

Additional Editor Comments:

Needs extensive language revisionUse consistent font styleHow do proper understanding of HF measured? Or you are speaking about the knowledge?Under your conclusion you have said, “…. patients with co-morbidities and depressive symptoms require extra care. What findings imply for such conclusion? Please rephrase it based on your findings.Write Pub Med as PubMedUse consistent bracket for references. E.g. you used square bracket for reference #24Please share us the data extraction checklist on Microsoft excel which you have used to extract data with the extracted data for both objectivesPlease correct the I^2  ^under the data analysis section on page 5Did you measured adherence to selfcare or the practice of good selfcare. Your result and what you have described in the methodology is confusing. Please clearly describe what was measured and how it was measured

Reviewers' comments:

Reviewer's Responses to Questions

**Comments to the Author**

1. Is the manuscript technically sound, and do the data support the conclusions?

Reviewer #1: Yes

Reviewer #2: Partly

2. Has the statistical analysis been performed appropriately and rigorously? 

Reviewer #1: Yes

Reviewer #2: Yes

3. Have the authors made all data underlying the findings in their manuscript fully available?

Reviewer #1: No

Reviewer #2: Yes

4. Is the manuscript presented in an intelligible fashion and written in standard English?

Reviewer #1: No

Reviewer #2: No

5. Review Comments to the Author

Reviewer #1: Independent Review Report

Evaluation

The evaluation is for the review article titled “Self-care Practices and associated factors among heart failure patients in Ethiopia: A systematic review and Meta-Analysis”

With the present study, the following areas are observed, commented and recommended.

1. Throughout the main text, it was sequential and well organized.

2. It requires grammatical corrections throughout the document.

3. This manuscript has a sum up of novelty.

Comments to the author/s

Abstract:

1. Try to avoid abbreviation of words in the abstract part

2. “We have used three databases such as Pub Med, Science Direct and Google Scholar.

Why do you exclude other important databases that you might get potential primary studies from?

3. Heterogeneity was evaluated using the I2 tests and the Cochrane Q test statistic.

What is the advantage of performing the Q test in addition to the I2 test to assess heterogeneity?

4. To examine publication bias, a funnel plot, Egger's weighted regression, and Begg's test were utilized. What is the importance of using different tests?

5. “The extent of self-care behaviour adherence is shown to be low among heart failure patients”. What is the standard reference to say “low or high”?

Method

6. Please use the uniform reference citation system

7. Please try to strictly adhere to all the PRISMA 2020 guidelines to write up your paper, especially the method section.

8. Does this review registered on PROSPERO?

9. What is your last search date included in the method part?

10. What you have done if a study has low quality or higher risk?

11. “Finfinnee" is not a legal name, not yet known

12. What you have done if there is heterogeneity and publication bias between the primary studies?

13. How to perform the pooled effect of odds ratio to identify the factors associated with good selfie care? Please show on your paper as it is critical.

Result

14. Table 1 column 6 gender (male). What is this???

15. Why you have performed sub-group analysis and sensitivity test?

Reviewer #2: Dear authors, thank you for your contribution. In general the issue you raised is very good. However, I have some comments and questions which described below.

1. Language edition service is required throughout the document

2. Please rephrase this sentence found in your abstract’ ‘The patient's understanding must therefore be improved by effective health education if self-care behaviour is to be improved.’’

3. The databases scrutinized to search studies are too small. Why only three databases? I think some papers are missed. Therefore please search on at least familiar databases like HINARI, Scopus, and AJOL

4. The MESH terms are little, why?

5. Why you preferred CoCoPop?

6. With this visible heterogeneity among studies, it is difficult to accept. Therefore, I need further evidence /explanation for this because the authors even mentioned the presence the presence of heterogeneity between studies conducted in Oromia. Why?

7. In your discussion part you mentioned this’ ’The prevalence of adherence is comparable with the study conducted in Brazil (35.17%)(9) and lower than the study of Netherland(41%)(6) and Taiwan(53.37%)(26) and higher than the study of Korea(31.98%)(14). This discrepancy could be due to the difference in the study method.’’ What type of difference in study method?

8. On your limitation please rephrase this sentence’ ‘The small sample due to a limited number of included studies with a small sample size for all included studies was a limitation of this study. ‘’

9. The number of studies included in your study is 11 but in the funnel plot it is more than 20 (each dot on the figure represent one study), Why this variation

Thank you

6. PLOS authors have the option to publish the peer review history of their article (what does this mean?). If published, this will include your full peer review and any attached files.

Reviewer #1: **Yes: **Mulugeta W/Selassie (Assistant Professor)

Reviewer #2: **Yes: **Bekahegn Girma Negie

---

## [Author Response · Author response to Decision Letter 0]

12 May 2023

Mohammed Feyisso Shaka, MPH

Academic Editor of PLOS ONE

Dear Editor of the Manuscript PONE-D-23-06552 Self-care practices and associated factors among heart failure patients in Ethiopia: A systematic review and Meta-Analysis," submitted to PLOS ONE. Thanks for your time and consideration in editing and reviewing the manuscript. We have carefully read your comments and corrected inline of your comments and suggestions. All comments raised were edited and incorporated in the revised manuscript. 

Here are the responses and elaborations for the comments from the editor and reviewer!

EDITOR COMMENTS

Editor comment: Needs extensive language revision

Author response: We have corrected the whole English grammar in the revised manuscript

Editor comment: Use consistent font style

Author response: We have used consistent font style in revised manuscript

Editor comment: How do proper understanding of HF measured? Or you are speaking about the knowledge?

Author response: We mean heart failure knowledge

Editor comment: Under your conclusion you have said, “…. patients with co-morbidities and depressive symptoms require extra care. What findings imply for such conclusion? Please rephrase it based on your findings.

Author response: Our result revealed the association between co-morbidities and depressive symptoms and we have rephrased it 

Editor comment: Write Pub Med as PubMed

Author response: We have corrected it throughout the documents

Editor comment: Use consistent bracket for references. E.g. you used square bracket for reference #24

Author response: We have corrected it in the documents

Editor comment: Please share us the data extraction checklist on Microsoft excel which you have used to extract data with the extracted data for both objectives

Author response: We will send it as supplementary file during submission

Editor comment: Please correct the I2 under the data analysis section on page 5

Author response: We have corrected it in the documents

Editor comment: Did you measured adherence to selfcare or the practice of good selfcare. Your result and what you have described in the methodology is confusing. Please clearly describe what was measured and how it was measured

Author response: We have measured the adherence level to self-care practice. The extent of adherence to self-care practice practices is low if less than the midpoint (50%) of the revised nine-item European Heart Failure Self-care Behavior Scale (EHFScBS-9)(25,26)

Editor comment: We noticed you have some minor occurrence of overlapping text with the following previous publication(s), which needs to be addressed:

Author response: We have paraphrased sentences to remove overlapping texts

REVIEWER COMMENTS

Reviewer 1:

Reviewer comments:. It requires grammatical corrections throughout the document. 

Author response: We have corrected the whole English grammar in the revised manuscript

Reviewer comments: Try to avoid abbreviation of words in the abstract part

Author response: We have omitted abbreviation under abstract

Reviewer comments: “We have used three databases such as Pub Med, Science Direct and Google Scholar.Why do you exclude other important databases that you might get potential primary studies from?

Author response: We have added the Scopus and other like HINARI are not freely accessed in our institution

Reviewer comments: Heterogeneity was evaluated using the I2 tests and the Cochrane Q test statistic. What is the advantage of performing the Q test in addition to the I2 test to assess heterogeneity?

Author response: The Q test identifies the Heterogeneity by using the Pvalue ≤ 0.05 and I2 statistics is used to assess the degrees of heterogeneity.

Reviewer comments: To examine publication bias, a funnel plot, Egger's weighted regression, and Begg's test were utilized. What is the importance of using different tests?

Author response: The funnel plot is the traditional diagrammatic representation used to assess the publication whereas, Egger's weighted regression, and Begg's test shows weather the publication bias was significant or not.

Reviewer comments: “The extent of self-care behaviour adherence is shown to be low among heart failure patients”. What is the standard reference to say “low or high”?

Author response: The extent of self-care practice practices is low if less than the midpoint (50%) of the revised nine-item European Heart Failure Self-care Behavior Scale (EHFScBS-9) as mentioned under outcome measurement

Reviewer comments: Please use the uniform reference citation system

Author response: We have used uniform reference citation system

Reviewer comments: Please try to strictly adhere to all the PRISMA 2020 guidelines to write up your paper, especially the method section.

Author response: We have modified the contents of method section as per PRISMA 2020 guidelines

Reviewer comments: Does this review registered on PROSPERO?

 Yes, it is already registered

Reviewer comments: What is your last search date included in the method part?

Author response: The last date to search was March 4, 2023.

Reviewer comments: What you have done if a study has low quality or higher risk?

Author response: Normally, a low quality studies was not found in our review. If the studies with a low quality are present, studies perceived to be of lower quality are removed and the analysis is then repeated.

Reviewer comments: “Finfinnee" is not a legal name, not yet known

Author response: “Finfinnee" is replaced by Addis Ababa

Reviewer comments: What you have done if there is heterogeneity and publication bias between the primary studies?

Author response: We will do a subgroup analysis and sensitivity analysis if heterogeneity and publication bias occurred:

Reviewer comments: How to perform the pooled effect of odds ratio to identify the factors associated with good selfie care? Please show on your paper as it is critical.

Author response: We have done by using the commands of The pooled effects of odds ratio was assessed by using the command of “metan logor selogor, xlab(0.1,1,10) label(namevar= authors) by ( factors)random texts(180) eform.”on STATA

Reviewer comments: Table 1 column 6 gender (male). What is this???

Author response: It represents the percentage of male participants in each study

Reviewer comments: Why you have performed sub-group analysis and sensitivity test?

Author response: Dear reviewer we haven’t done sub-group analysis and sensitivity test because our study was not heterogeneous

Reviewer 2:

Reviewer comments: Language edition service is required throughout the document

Author response: We have corrected the whole English grammar in the revised manuscript

Reviewer comments: Please rephrase this sentence found in your abstract’ ‘The patient's understanding must therefore be improved by effective health education if self-care behaviour is to be improved.’’

Author response: We have rephrased it in revised manuscript

Reviewer comments: The databases scrutinized to search studies are too small. Why only three databases? I think some papers are missed. Therefore please search on at least familiar databases like HINARI, Scopus, and AJOL

Author response: We have added the Scopus and other like HINARI are not freely accessed in our institution

Reviewer comments: The MESH terms are little, why?

Author response: Dear reviewer, the MESH term is sufficient for obtain our outcome variables

Reviewer comments: Why you preferred CoCoPop?

Author response: 

The CoCoPop framework can be used for reviews addressing a question relevant to prevalence or incidence

The CoCoPop framework is used for research questions focusing on associated factors. It is used most frequently to determine associations between certain risk factors or exposures and an outcome. 

Reviewer comments: With this visible heterogeneity among studies, it is difficult to accept. Therefore, I need further evidence /explanation for this because the authors even mentioned the presence the presence of heterogeneity between studies conducted in Oromia. Why?

Author response: Generally, there was no heterogeneity in our study. The variations seen in Oromia might be due to a difference in publication years and large variations in sample size.

Reviewer comments: In your discussion part you mentioned this’ ’The prevalence of adherence is comparable with the study conducted in Brazil (35.17%)(9) and lower than the study of Netherland(41%)(6) and Taiwan(53.37%)(26) and higher than the study of Korea(31.98%)(14). This discrepancy could be due to the difference in the study method.’’ What type of difference in study method?

Author response: As mentioned in discussion, this discrepancy could be due to the difference in the sample size and sampling techniques and different data collection used.

Reviewer comments: On your limitation please rephrase this sentence’ ‘The small sample due to a limited number of included studies with a small sample size for all included studies was a limitation of this study. ‘’

Author response: We have rephrased it as per your recommendation

Reviewer comments: he number of studies included in your study is 11 but in the funnel plot it is more than 20 (each dot on the figure represent one study), Why this variation

Author response: To assess the publication bias we have used the commands for the Odds ratio/associated factors. Therefore, the number of the dots represents the number of associated factors. 

Dear reviewer, the number of dots represents the number of studies if the publication bias was depicted by using the commands for the magnitude of the problem.

Thanks for your time and consideration,

 Regards!

---

## [Editor Report · Decision Letter 1]

5 Jul 2023

Adherence to self-care practices and associated factors among heart failure patients in Ethiopia: A systematic review and Meta-Analysis

PONE-D-23-06552R1

Dear Dr. Bekele,

We’re pleased to inform you that your manuscript has been judged scientifically suitable for publication and will be formally accepted for publication once it meets all outstanding technical requirements.

Kind regards,

Mohammed Feyisso Shaka, MPH

Academic Editor

PLOS ONE
---

## [Editor Report · Acceptance letter]

11 Aug 2023

PONE-D-23-06552R1 

Adherence to self-care practices and associated factors among heart failure patients in Ethiopia: A systematic review and meta-analysis 

Dear Dr. Bekele:

I'm pleased to inform you that your manuscript has been deemed suitable for publication in PLOS ONE. Congratulations! Your manuscript is now with our production department. 

Kind regards, 

on behalf of

Mr. Mohammed Feyisso Shaka 

Academic Editor

PLOS ONE